# Obscured inequity:How focusing on rates of disparities can conceal inequities in the reasons why adolescents are unvaccinated

Elizabeth M. Anderson*

Department of Sociology, Indiana University, Bloomington, Indiana, United States of America

* anderelm@iu.edu

## Abstract

Traditional sociodemographic disparities in adolescent vaccination initiation for the HPV, Tdap, and MenACWY vaccines have declined in the United States of America. This decline raises the question of whether inequities in access have been successfully addressed. This paper synthesizes research on the resource barriers that inhibit vaccination alongside research on vaccine hesitancy where parents actively refuse vaccination. To do so, I classify the primary reason why teens are unvaccinated in the National Immunization Survey-Teen 2012–2022 into three categories: resource failure, agentic refusal, and other reasons. I use three non-exclusive subsamples of teens who are unvaccinated against the HPV ($N$ = 87,163), MenACWY ($N$ = 54,726), and Tdap ($N$ = 10,947) vaccines to examine the relative importance of resource failure reasons and agentic refusal reasons for non-vaccination across time and teens' sociodemographic characteristics. Results indicate that resource failure reasons continue to explain a substantial portion of the reasons why teens are unvaccinated and disproportionately affect racially/ethnically and economically marginalized teens. Thus, even as sociodemographic inequalities in *rates* of vaccination have declined, inequities in access remain consequential.

**Data Availability Statement:** The data are publicly available on the United State's Centers for Disease Control website. Data from 2012-2014: https://www.cdc.gov/vaccines/imz-managers/nis/datasets-teen.html#2008-2014 Data from 2015-

## Introduction

Oftentimes, when quantitative scholars identify health-based inequalities, they begin by analyzing disparities in rates of outcomes by social characteristics like gender, race/ethnicity, and socioeconomic status. Because disparities in rates are tangible and easy to understand, they provide clear targets for policy makers, public health professionals, and scholars to investigate both how these inequalities are created and potential interventions to address these inequalities. Moreover, persistent health-based disparities are often explained by resource disparities, as less advantaged groups have access to fewer flexible resources, which would allow them to avoid emergent health risks [1–4].

Given the remarkably durable association between resource inequalities and healthcare access/health outcomes, scholars must investigate whether the absence of disparities is truly a result of ameliorated inequalities. This question is particularly relevant in the case of the

2020: https://www.cdc.gov/vaccines/imz-managers/nis/datasets-teen.html

**Funding:** The author received no specific funding for this work.

**Competing interests:** The author has declared that no competing interests exist.

vaccines recommended for adolescents in the United States of America: the HPV (human papillomavirus), Tdap (tetanus, diphtheria, and pertussis) and MenACWY (meningococcal conjugate) vaccines [5]. As gender, racial/ethnic, and income disparities in the initiation of these vaccines have gradually narrowed or even reversed [6], some scholars have begun to investigate whether "reverse disparities" in vaccination are emerging [7].

While there are substantial bodies of literature focusing on why children are unvaccinated, they tend to take two disparate approaches by focusing on either vaccine hesitancy or undervaccination. The World Health Organization (WHO) defines vaccine hesitancy as "delay in acceptance or refusal of vaccination despite availability of vaccination services" [8]. When vaccine hesitancy is operationalized in research, this research tends to focus on advantaged parent's agentic use of resources to avoid vaccinating their children [9–12]. Conversely, health disparities scholars who study inequities in vaccination, refer to children as "undervaccinated" when they are missing at least one childhood vaccine because of the unavailability or inaccessibility of vaccine(s) due to resource deficits as a result of structural disadvantage [13, 14].

Vaccination is a global challenge, as counties across the world face varying levels of obstacles in both providing access to vaccines as well as addressing a growing tide of vaccine hesitancy [15–17]. Sociodemographic disparities in vaccination levels are present across many countries, as structurally disadvantaged communities frequently access vaccines at lower levels [e.g., 18, 19]. In Canada, scholars have noted that overall high levels of vaccination nonetheless have the potential to conceal important sociodemographic disparities in vaccination levels [20]. Although vaccine hesitancy manifests differently across countries, the growing rise of misinformation and vaccine hesitancy has garnered international attention, with the WHO highlighting vaccine hesitancy as one of the top ten threats to global health [11, 15]. Nevertheless, most studies of vaccination globally tend to consider vaccine hesitancy or undervaccination separately.

This project integrates these two frameworks to investigate whether the equalizing rates of adolescent vaccination initiation against the HPV, Tdap, and MenACWY vaccines is indeed indicative of decreasing disparities in access to vaccination. To address this question, I use three non-exclusive subsamples of the National Immunization Survey-Teen from 2012–2020, which capture the reasons why teens are unvaccinated against the HPV, MenACWY, and Tdap vaccines. I classify the reasons why teens are unvaccinated into three categories: those associated with agentic refusal (consistent with research on vaccine hesitancy), resource failure (consistent with research on undervaccination), and "other" reasons. I estimate three multinomial logistic models predicting the reason classification in the HPV, MenACWY, and Tdap samples. Each model examines how the relative importance of agentic refusal and resource failure shifts over time, by sociodemographic characteristics, and by adolescents' overall vaccination status. Results indicate the persistent, but uneven, importance of failed resources in the reasons why teens are unvaccinated, even as privileged parents increasingly leverage their resources to avoid vaccinating their children.

## Background

The unequal distribution of economic, social, and cultural resources across society is associated with consequential inequalities in healthcare access as higher-resourced groups can leverage their resources to avoid health threats [2–4, 21]. While the influence of some forms of resources, for example economic capital to purchase health insurance and pay for copays, are obvious, other subtler forms of resources are also important. Patients who can draw upon less visible resources, such as cultural health capital [interpersonal skills and resources that facilitate interactions with healthcare providers, 22] have greater access to specialized and

sophisticated care [23, 24]. Importantly, these resources are also flexible so they can adapt over time, which results in persistent disparities even as policymakers, healthcare providers, and public health officials act to address the mechanisms that generate inequities.

Alternatively, a substantial body of research on disparities in vaccination explains sociodemographic disparities through a resource deficit perspective (i.e., undervaccination). Under the sociodemographic disparities framework, undervaccination occurs when parents cannot leverage the necessary economic, social, or cultural health resources to access vaccination [13, 25, 26]. The Vaccines for Children (VFC) program finances vaccines for children who would otherwise be unable to afford them [27]. While this program addresses disparities caused by low economic resources and successfully decreased many income-based disparities in childhood vaccination, disparities in vaccination have persisted because economic resources are not the only resource that must be leveraged to access vaccines [28, 29]. These resources can include lack of a provider recommendation [26], limited access to transportation [30], accessibility of healthcare [25], and low information [31].

On the opposite end of the spectrum, there is convincing evidence that some privileged parents actively refuse vaccination [i.e., vaccine hesitancy, 9, 10, 14, 17, 32]. Research on vaccine hesitancy reveals how parents' individualistic ideologies shape decisions to delay or refuse childhood vaccines and the resources that parents deploy to avoid vaccination [9–12, 33–35]. There are several reasons why parents actively decide against vaccinating their children. Some parents are motivated to reject vaccines due to a distrust of doctors, public health officials, the scientific testing process, and/or the regulatory system [32, 36, 37]. Other parents reject vaccines due to the perceived negative side effects of previous vaccines that their children received [9]. Furthermore, parents engaged in intensive forms of parenting report believing that vaccines are not needed because of a low perceived risk of infection and their ability to engage in alternative forms of prevention through breastfeeding, diet, reducing exposure to chemicals, among others [32, 38, 39].

Because Americans are generally supportive of childhood vaccines, parents who refuse all or some vaccines often have to leverage economic, social, and cultural health resources to avoid vaccination [9, 34, 40]. This could include finding pediatricians who will support their vaccination decisions [even if the physician does not accept health insurance, 9, 34], applying for personal belief exemptions [12], selecting into schools with lower levels of vaccination [10, 32, 41], and using religious beliefs to receive exemptions [42]. Consequently, the decision to decline vaccination has increasingly been considered a choice that is enabled by privilege [14, 43].

The disappearance of traditional sociodemographic disparities in adolescent vaccination initiation raises the question of whether resource failures or agentic refusal patterns are the primary reason why teens are unvaccinated [6]. Traditionally, undervaccination and vaccine hesitancy are studied separately. Consequently, we do not know which explanation for non-vaccination predominates. This study draws upon both bodies of literature to classify a broad spectrum of reasons for non-vaccination into reasons associated with resource failure and activation and their relative importance over time and across adolescents' sociodemographic characteristics. If agentic refusal reasons predominate as the primary reason why teens are unvaccinated, this finding would point to the success of public health and clinical interventions to reduce barriers to initiating vaccination. However, if resource failures predominate, this study would illustrate how a focus on utilization disparities could obscure the persistent role of resources in generating inequalities.

**Adolescent vaccines.** Although adolescent vaccines have received comparatively less attention in research on the reasons why children are unvaccinated, they provide a unique opportunity to evaluate the relative importance of inequalities in resources to patterns of

vaccination. Each of the three vaccines considered here has the potential to protect adolescents against different risks and are embedded in unique social histories.

The human papillomavirus virus is a sexually transmitted infection, and an estimated 42.5% of non-institutionalized American adults have some type of HPV [44]. Although most cases of HPV resolve on their own without medical intervention, HPV is associated with cancers of the cervix, vagina, vulva, penis, anus, rectum, and oropharynx, slightly less than half of which occur in males [45]. Sexual politics infused the public discourse surrounding the HPV vaccine, and the first version of the vaccine, Gardasil[®], was called the "promiscuity vaccine" by some conservatives, because of the vaccine's ability to protect against a sexually transmitted infection [46]. Early state mandates requiring the HPV vaccine for school attendance faced strong backlash, and, in 2020, only three states had laws requiring the HPV vaccine for school attendance. Rates of HPV vaccination lag behind those for the MenACWY and Tdap vaccines, even as the initial sociodemographic disparities in HPV vaccination lessened over time [6, 47]. In 2021, racial and ethnic minority teens were as or more likely to receive at least one dose of the HPV vaccine compared to non-Hispanic White teens–a reversal of initial disparities [6, 47].

The MenACWY and Tdap vaccines do not have the same contentious social history as the HPV vaccine. The MenACWY vaccine protects against four strains of the meningococcal bacteria, which cause meningitis, an infection of the brain and spinal cord that can lead to disability or death [48]. The Tdap vaccine protects against three diseases; two of which can be spread person to person (diphtheria and pertussis) and one that is spread via cuts or puncture wounds (tetanus) [49]. Whereas parents who refuse other vaccinations for their children can draw upon the assumption that community-level or individual health behaviors will protect their children from contracting a communicable disease, tetanus is unique because it is not spread through interpersonal contact [9, 38]. Consequently, previous research has noted that parents who otherwise refuse vaccinations for their children, may elect to vaccinate their children against tetanus [9]. In 2020, 30 states and the District of Columbia had laws requiring at least one dose of the MenACWY vaccine for school attendance, and 49 states and the District of Columbia had laws requiring at least one dose of the Tdap vaccine. The MenACWY and Tdap vaccines were not plagued with similar widespread racial/ethnic and economic disparities as those found in earlier years of the HPV vaccine rollout [50].

## Materials & methods

### National Immunization Survey-Teen

This paper uses pooled cross-sectional data from the publicly available National Immunization Survey-Teen (NIS-Teen) from 2012 to 2020. This survey, collected annually by the Centers for Disease Control (CDC) starting in 2008, identifies households with an adolescent aged 13–17 and interviews a parent or guardian about the adolescent's immunization status [51]. The NIS-Teen interviewers verbally obtained informed consent from the interviewee (the teen's parent or guardian) at the beginning of the audio-recorded interview. Because the publicly available version of the NIS-Teen is de-identified and publicly available, the present study is not considered human subjects research and does not require review by an institutional review board.

Because the CDC recommends that all teens receive the HPV, MenACWY, and Tdap vaccines between 11–12 years of age, this sample captures adolescents during the "catch-up" period for vaccination [5]. Comprehensive information on the survey instrument can be found in in NIS-Teen documentation [51]. The present analyses exclude adolescents living in U.S. territories ($n$ = 6,838 of the total pooled NIS-Teen sample of 364,129 [1.9%]). Although the first wave of the NIS-Teen was collected in 2008, I begin the sample with NIS-Teen data

from 2012, when the CDC universally recommended all three vaccines to all adolescents, so that all parents in the sample were making decisions under the same CDC guidelines [52, 53].

## Analytic sample

All analyses are conducted on non-exclusive samples (e.g., teens unvaccinated against the HPV, MenACWY, and Tdap vaccines are included in all three analytic samples): (1) adolescents who are unvaccinated or partially vaccinated against HPV (hereafter referred to as "teens who are unvaccinated against HPV" for simplicity), (2) adolescents unvaccinated against MenACWY, and (3) adolescents unvaccinated against Tdap. The skip logic for which parents were asked why their teen was unvaccinated differed slightly between vaccine types. Parents who reported that their child had not received any doses of the MenACWY and/or Tdap vaccine were asked for the main reason for not vaccinating. However, only parents who reported that their child had received no doses of the HPV vaccine or fewer than the recommended doses of the HPV vaccine *and* answered that the adolescent was not likely to be vaccinated in the next year were asked for the main reason why their teen was unvaccinated against HPV.

## Measures

**Dependent variables.** I constructed three dependent variables that correspond to the three analytic samples that capture parents' reason for why their teens are unvaccinated against each vaccine type. In the publicly available data, the NIS-Teen recodes the parent's (or guardian's) answer to the question of why their teen is not vaccinated to identify the main reason, and this answer is represented as a series of binary variables. I classified reasons into three categories: reasons associated with parents' failure to activate resources to vaccinate their children ("resource failure"), reasons associated with parents' intentional decisions to avoid vaccination ("agentic refusal"), and other reasons that did not fit in the prior categories (see Table 1). The reasons categorized in the resource failure category aligns with previous literature on the barriers to vaccination for low-resourced parents. In these cases, parents were unable to access the necessary resources to overcome these barriers to vaccination. Reasons placed in the agentic refusal category are drawn from the literature on non-vaccination. Although the WHO includes "convenience" within their definition of vaccine hesitancy, I follow the guidance of Bedford and colleagues [54] to exclude reasons most closely associated with convenience from the agentic refusal category (e.g., "intend to but have not yet"), because of the potential for convenience to blur the line between structural barriers to vaccination and individual-level decisions.

The categories of the reasons why teens were unvaccinated were largely consistent across the three vaccines, and additional categories were added between years as new reasons emerged. Across the sample frame, there were 29 reason categories for why teens were unvaccinated against HPV, and 24 reasons for why teens were unvaccinated against MenACWY and Tdap. Between 2012–2020, 127,820 respondents answered the question of why their teen was not vaccinated/not fully vaccinated against HPV, 62,411 respondents answered the question of why their teen was not vaccinated against MenACWY, and 13,100 respondents answered the question of why their teen was not vaccinated against Tdap.

The overwhelming majority of respondents gave only one reason why their teen was unvaccinated. Within each sample, between 0.2% and 0.3% of parents of teens unvaccinated against HPV, MenACWY, and/or Tdap gave more than one reason. After constructing the dependent variables, I identified how many respondents who described multiple reasons for why their teen was unvaccinated reported reasons that fell within both the agentic refusal or failure

**Table 1. Classification of reasons for teen's unvaccinated status, NIS-Teen 2012–2020.**

| Resource failure reasons | Agentic refusal reasons | Other reasons |
|---|---|---|
| • Provider did not recommend vaccination<br>• Cost prohibitive<br>• Vaccine is not available in healthcare provider's office<br>• No doctor/ doctor visit not scheduled<br>• Difficulty making/getting to appt/ transportation problem<br>• Time prohibitive<br>• Did not know about the diseases/did not know was recommended for their teen<br>• Need more information/new vaccine concerns[†]<br>• COVID-19 pandemic<br>* Teen is male | • Doesn't believe in vaccines<br>• Family/parent decision not to vaccinate<br>• Safety concerns<br>• Effectiveness concern<br>• Teen should make decision<br>• Teen is fearful<br>• Religious reason<br>• Not needed/not necessary<br>* Increased sexual activity concern | • Already up to date<br>• Not a school requirement<br>• Teen is not the appropriate age/provider indicated could vaccinate at older age<br>• College shot<br>• Intend to but have not yet<br>• Handicapped/ special needs/ illness precludes vaccination<br>• Other reason<br>• Teen is already sexually active<br>• Teen is not sexually active<br>• Not getting HPV vaccine because teen does not have an ob-gyn. |

* Indicates that this reason is only included in the HPV un-vaccinated sample.

[†]These reasons were classified as resource failures because they potentially indicate low knowledge about the vaccine. Low knowledge is generally understood to be a fundamental cause of health disparities [21]. Consequently, lower knowledge could suggest that parents are unable to access the social or cultural health capital needed to obtain greater information about the vaccine. However, some parents who refuse vaccines for their children justify their decisions because of a need for more information [39]. To ensure that my findings are robust to the classification of this indicator, I conducted sensitivity analyses that reassigned the "need more information" reason to the agentic refusal category. The results were substantively identical, indicating that the main findings are robust to this alternative specifications (see S1 Table).

categories, and I dropped these cases from analyses ($n = 133$ for HPV sample; $n = 53$ for MenACWY sample; $n = 10$ for Tdap sample).

*Independent variables.* The analyses included controls for several measures of adolescent characteristics, including teen's sex (0 = female, 1 = male), adolescent's race/ethnicity (0 = non-Hispanic White, 1 = Hispanic, 2 = Black, and 3 = multiracial/other race), teen's age (13–17 years), family income (2 = below poverty, 1 = above poverty but below \$75,000, and 0 = above \$75,000), maternal education (1 = less than high school, 0 = high school degree, 2 = some college, and 3 = college degree), and Census region (0 = Northeast, 1 = Midwest, 2 = South, 3 = West).

I also include a vaccination history measure of which vaccines the adolescent is unvaccinated against to capture the teens that overlap between the samples. These variables have four categories: (1) only unvaccinated against the primary vaccine considered, (2) unvaccinated against the primary vaccine and one of the other vaccines, (3) unvaccinated against the primary vaccine and the other vaccine not included in category 2, and (4) unvaccinated against all three vaccines.

Lastly, I constructed a measure of state-level mandates for vaccination by grade and teen's sex. Starting from the list of state-level mandates for the HPV, MenACWY, and Tdap vaccine found at immunize.org, I validated the year of each mandate implementation by searching for local media articles, state health department news releases, or school district fliers to confirm the year of implementation. I then constructed a measure that identified which school grades had vaccine mandates between 2012–2020. Detailed information about the construction of this measure can be found in S1 File, and the measure can be found in S2 File. For these analyses, I created three binary variables to indicate whether or not each respondent in the analytic sample was under a vaccine mandate for the HPV, MenACWY, or Tdap vaccines based on their age, sex, and state of residence.

## Statistical analyses

All analyses were conducted in Stata 17.1. To examine differences between respondents unvaccinated due to agentic refusal or failure reasons, I estimate cross-tabulations and weighted proportions to describe the samples. I then estimate three multinomial logistic regression models with survey weights. Every model includes the survey year, teen's sex, teen's race/ethnicity, teen's age, family income, mother's education, census region of residence, the vaccination history variables, and state mandate variables. Each model includes a quadratic specification for survey year, and the model predicting why teens are unvaccinated against HPV includes an interaction between adolescent's sex, year, and the quadratic term for year.

I present the regression results in the probability metric with average marginal effects (AME) to aid interpretation. Because of the large sample size, I present 95 percent confidence intervals and p-values; although I only discuss the confidence intervals as the use of p-values in large sample size research tends toward zero even when the magnitude of the effect is relatively small [55]. Average marginal effects for the "other" reasons why teens are unvaccinated and odds ratios for the models can be found in S2 Table.

All survey weights were constructed following the NIS-Teen guidelines for combining multiple years of data [51]. The public-use data file provides data on teens' sex and race as well as maternal education that has already been imputed using a sequential hot-deck method (see (44) for detailed information). NIS-Teen does not impute missing values for family income ($n = 6,380$ or 6.7% of the HPV sample; $n = 2,481$ or 5.0% of the MenACWY sample; $n = 642$ or 5.9% of the Tdap sample). Additionally, some teens did not have vaccination status information on all three vaccines ($n = 26,152$ or 27.7% of the HPV sample; $n = 3,824$ or 7.8% of the MenACWY sample; $n = 1,969$ or 18.2% of the Tdap sample). Cases with missing values were dropped using listwise deletion.

## Results

The descriptive results reveal several interesting patterns (see Table 2). First, the total sample of teens who are unvaccinated against HPV ($N = 87,163$) is greater compared to those teens who have not received the MenACWY vaccine ($N = 54,726$), and the Tdap vaccine ($N = 10,947$). While the greatest proportion of teens in the HPV sample were unvaccinated because of agentic refusal reasons compared to resource failure reasons (40% vs. 34%), teens in the MenACWY and Tdap samples were more likely to be unvaccinated as a result of resource failures versus agentic refusal reasons (57% vs. 23% in the MenACWY sample; 44% vs 34% in the Tdap sample. In general, the number of unvaccinated teens in each sample decreased over time, which is consistent with the previously identified increase in adolescent vaccination over time [6]. There are more males than females in the HPV sample (55% male) and slightly more males than females in the MenACWY sample (51% male), and no notable gender imbalance in the Tdap sample (50% male).

There were also noteworthy patterns based on the characteristics of the teen and their family. While around three-fifths of the teens in the HPV and MenACWY samples were White, only about half of all teens in the Tdap sample were White. Compared to the HPV and MenACWY samples, a higher proportion of teens in the Tdap sample were Hispanic or Black. While around 43% of teens in the HPV and MenACWY samples had mothers with college degrees, only around 35% of teens in the Tdap sample had mothers with a college degree. Furthermore, around 45% of teens in the HPV and MenACWY sample lived in households with family incomes greater than $75,000, but only 36% of teens in the Tdap sample lived in households with incomes greater than $75,000. When comparing each of the samples, the highest number of teens were unvaccinated only against HPV (meaning that they were vaccinated

**Table 2. Descriptive statistics for the HPV, MenACWY, and Tdap samples of unvaccinated adolescents, National Immunization Survey, Teen 2012–2022.**

| | HPV | | MenACWY | | Tdap | |
| | (N = 87,163) | | (N = 54,726) | | (N = 10,947) | |
| *Categorical Variables* | *n* | *Percent[a]* | *n* | *Percent[a]* | *n* | *Percent[a]* |
|---|---|---|---|---|---|---|
| Reason not vaccinated | | | | | | |
| Agentic refusal | 35,310 | 40.14% | 12,658 | 23.38% | 3,691 | 34.11% |
| Resource failure | 28,845 | 34.30% | 30,536 | 56.65% | 4,668 | 44.46% |
| Other | 23,008 | 25.56% | 11,532 | 19.98% | 2,588 | 21.43% |
| Year[1] | | | | | | |
| 2012 | 11,146 | 15.30% | 10,038 | 21.69% | 2,308 | 23.73% |
| 2013 | 9,866 | 13.51% | 9,246 | 19.77% | 1,840 | 19.52% |
| 2014 | 9,671 | 11.34% | 6,054 | 10.75% | 1,027 | 9.87% |
| 2015 | 10,407 | 10.77% | 5,845 | 9.32% | 884 | 7.28% |
| 2016 | 9,765 | 10.50% | 5,467 | 8.98% | 1,164 | 9.76% |
| 2017 | 9,898 | 10.07% | 5,290 | 8.04% | 1,091 | 8.48% |
| 2018 | 8,943 | 10.31% | 4,321 | 7.64% | 949 | 7.98% |
| 2019 | 8,430 | 8.94% | 4,574 | 7.51% | 979 | 7.61% |
| 2020 | 9,037 | 9.28% | 3,891 | 6.31% | 705 | 5.78% |
| Adolescent's sex | | | | | | |
| Female | 37,874 | 44.58% | 26,514 | 48.99% | 5,408 | 50.27% |
| Male | 49,289 | 55.42% | 28,212 | 51.01% | 5,539 | 49.73% |
| Adolescent's age[1] | | | | | | |
| 13 | 16,368 | 18.56% | 10,700 | 19.82% | 2,118 | 19.07% |
| 14 | 16,805 | 19.35% | 10,994 | 19.68% | 2,047 | 18.16% |
| 15 | 17,842 | 20.31% | 11,271 | 20.27% | 2,128 | 20.02% |
| 16 | 18,496 | 20.94% | 11,392 | 20.90% | 2,330 | 20.82% |
| 17 | 17,652 | 20.83% | 10,369 | 19.33% | 2,324 | 21.93% |
| Adolescent's race/ ethnicity | | | | | | |
| White | 59,481 | 61.26% | 38,303 | 62.62% | 6,579 | 53.11% |
| Hispanic | 11,260 | 16.93% | 6,810 | 16.32% | 1,941 | 21.66% |
| Black | 7,461 | 12.25% | 4,358 | 12.42% | 1,289 | 17.10% |
| Multiracial/other race | 8,961 | 9.57% | 5,255 | 8.65% | 1,138 | 8.12% |
| Family income | | | | | | |
| > $75,000 | 45,005 | 45.67% | 28,362 | 44.94% | 4,690 | 36.09% |
| Above poverty ≤ $75,000 | 31,956 | 37.85% | 20,022 | 38.49% | 4,321 | 39.56% |
| Below poverty | 10,202 | 16.49% | 6,342 | 16.56% | 1,936 | 24.35% |
| Mother's Education | | | | | | |
| Less than high school | 5,735 | 8.16% | 3,607 | 8.28% | 1,300 | 13.67% |
| High school | 13,698 | 21.26% | 8,449 | 21.03% | 2,157 | 25.42% |
| Some college | 25,758 | 28.04% | 16,041 | 27.86% | 3,080 | 25.96% |
| College degree | 41,972 | 42.55% | 26,629 | 42.83% | 4,410 | 34.94% |
| Census Region | | | | | | |
| Northeast | 14,220 | 14.91% | 8,783 | 14.06% | 1,797 | 15.61% |
| Midwest | 19,402 | 22.89% | 12,551 | 23.83% | 2,346 | 22.47% |
| South | 32,893 | 38.80% | 19,644 | 38.82% | 4,242 | 41.04% |
| West | 20,648 | 23.40% | 13,748 | 23.29% | 2,562 | 20.88% |
| Not vaccinated against. . . | | | | | | |
| HPV only | 52,707 | 60.02% | -- | -- | -- | -- |
| MenACWY only | -- | -- | 13,384 | 21.45% | -- | -- |

*(Continued)*

**Table 2.** (Continued)

| Categorical Variables | HPV (N = 87,163) | | MenACWY (N = 54,726) | | Tdap (N = 10,947) | |
|---|---|---|---|---|---|---|
| | n | Percent[a] | n | Percent[a] | n | Percent[a] |
| Tdap only | -- | -- | -- | -- | 1,177 | 10.56% |
| HPV & Tdap | 1,188 | 1.39% | -- | -- | 1,683 | 16.63% |
| HPV & MenACWY | 17,886 | 18.46% | 18,335 | 30.17% | -- | -- |
| MenACWY & Tdap | -- | -- | 5,572 | 11.68% | 1,857 | 15.73% |
| HPV, MenACWY & Tdap | 15,382 | 20.13% | 17,435 | 36.70% | 6,230 | 57.09% |
| State vaccine mandate | | | | | | |
| Yes | 85,494 | 98.89% | 41,503 | 77.71% | 4,348 | 37.17% |
| No | 1,669 | 1.11% | 13,223 | 22.29% | 6,599 | 62.83% |

[a]Percentages are weighted. Respondents may overlap between samples.

[b]Although year and age are included as continuous variables in the regression analyses, I present the descriptive results for each individual year and age.

with the MenACWY and Tdap vaccines, $n$ = 52,707), 13,384 teens were missing only the MenACWY vaccine, and 1,177 teens were missing only the Tdap vaccine. Intriguingly, almost three fifths (57%) of teens in the Tdap sample were also missing the other two adolescent vaccines, while only 37% of teens in the MenACWY sample were also missing the other two vaccines, and only 20% of teens in the HPV sample were also missing the other two vaccines.

Table 3 presents the results of three multinomial logistic regression models predicting agentic refusal and failure reasons as the primary reason why teens in the HPV, MenACWY, and Tdap samples were unvaccinated. Results are presented as average marginal effects with 95% confidence intervals. Teens in the HPV sample had the lowest predicted probability of being unvaccinated due to resource failure reasons compared to teens unvaccinated against MenACWY and Tdap samples. The predicted probability of being unvaccinated against HPV because of resource failure reasons was 0.34 compared to the 0.40 predicted probability of being unvaccinated because of agentic refusal reasons. The predicted probability of teens being unvaccinated due to resource failure reasons was 0.44 for teens in the Tdap sample, compared to a 0.34 predicted probability of being unvaccinated due to agentic refusal reasons. Meanwhile, teens in the MenACWY sample had the highest predicted probability of being unvaccinated due to resource failure reasons (0.57) compared to a 0.23 predicted probability of being unvaccinated as a result of agentic refusal reasons.

The association between year and the reasons why teens are unvaccinated differed by vaccine type and are most clearly visible graphically (see Fig 1). In 2012, teens had a higher predicted probability of being unvaccinated due to resource failure reasons compared to agentic refusal reasons in each sample. Although the predicted probabilities follow distinct patterns in each of the three samples, the predicted probabilities of agentic refusal and resource failure reasons converged between 2012 and 2020 within each sample. While predicted probability of being unvaccinated due to resource failure reasons decreased across each sample, in 2020, only teens in the MenACWY sample were still more likely to be unvaccinated as a result of resource failures reasons compared to agentic refusal reasons. For each sample, the lowest predicted probability of being unvaccinated as a result of resource failure reasons occurred in 2018, where the predicted probability was 0.28 for the HPV sample, 0.46 for the MenACWY sample, and 0.32 for the Tdap sample.

The sociodemographic characteristics of teens did not have a consistent association with the reason for being unvaccinated across vaccine types. Across all samples, males were less

**Table 3. Average marginal effects for the multinomial logistic regressions predicting agentic refusal and failure reasons for teen's unvaccinated status for the HPV, MenACWY, and Tdap samples, NIS-Teen 2012–2020.**

| | HPV AME (95% CI) | | MenACWY AME (95% CI) | | Tdap AME (95% CI) | |
|---|---|---|---|---|---|---|
| | *Agentic Refusal* | *Resource Failure* | *Agentic Refusal* | *Resource Failure* | *Agentic Refusal* | *Resource Failure* |
| Year | 0.00 | -0.02[***] | 0.03[***] | -0.03[***] | 0.03[***] | -0.04[***] |
| | (0.00, 0.00) | (-0.02, -0.01) | (0.03, 0.03) | (-0.04, -0.03) | (0.02, 0.04) | (-0.05, -0.04) |
| Male *vs. female/ other gender* | -0.07[***] | 0.09[***] | -0.03[***] | 0.03[***] | -0.04[*] | 0.01 |
| | (-0.08, -0.06) | (0.08, 0.10) | (-0.04, -0.02) | (0.02, 0.05) | (-0.07, -0.01) | (-0.02, 0.04) |
| Age | 0.01[**] | 0.00 | 0.01[***] | -0.01[***] | 0.01 | -0.01[*] |
| | (0.00, 0.01) | (-0.01, 0.00) | (0.00, 0.01) | (-0.02, 0.00) | (0.00, 0.02) | (-0.02, 0.00) |
| Teen's race/ethnicity | | | | | | |
| Hispanic *vs. White* | -0.04[***] | 0.05[***] | -0.02 | 0.02 | -0.04 | 0.09[***] |
| | (-0.06, -0.02) | (0.03, 0.07) | (-0.04, 0.01) | (-0.01, 0.04) | (-0.09, 0.00) | (0.04, 0.14) |
| Black *vs. White* | -0.04[***] | 0.04[***] | 0.02 | -0.01 | 0.02 | 0.01 |
| | (-0.06, -0.02) | (0.02, 0.06) | (-0.01, 0.04) | (-0.03, 0.02) | (-0.02, 0.07) | (-0.03, 0.05) |
| Multiracial/other race *vs. White* | -0.05[***] | 0.06[***] | -0.01 | 0.03[*] | -0.01 | 0.06[*] |
| | (-0.07, -0.03) | (0.04, 0.08) | (-0.03, 0.02) | (0.01, 0.06) | (-0.06, 0.04) | (0.00, 0.11) |
| Hispanic *vs. Black* | 0.00 | 0.01 | -0.03[*] | 0.02 | -0.06[*] | 0.08[**] |
| | (-0.02, 0.03) | (-0.02, 0.01) | (-0.06, 0.00) | (-0.01, 0.06) | (-0.12, -0.01) | (0.02, 0.14) |
| Hispanic *vs. Multiracial/other race* | 0.02 | -0.01 | -0.01 | -0.01 | -0.03 | 0.03 |
| | (-0.01, 0.04) | (-0.04, 0.01) | (-0.04, 0.02) | (-0.05, 0.02) | (-0.09, 0.03) | (-0.03, 0.10) |
| Black *vs. Multiracial/other race* | 0.01 | -0.02 | 0.02 | -0.04[*] | 0.03 | -0.05 |
| | (-0.01, 0.04) | (-0.05, 0.01) | (-0.01, 0.05) | (-0.07, 0.00) | (-0.04, 0.09) | (-0.11, 0.02) |
| Family income | | | | | | |
| Below poverty *vs. > $75,000* | -0.03[*] | 0.05[***] | 0.01 | 0.03 | -0.04 | 0.05 |
| | (-0.05, 0.00) | (0.03, 0.07) | (-0.01, 0.04) | (0.00, 0.05) | (-0.09, 0.01) | (0.00, 0.10) |
| Below poverty *vs. above poverty ≤ $75,000* | -0.04[***] | 0.06[***] | 0.00 | 0.03[*] | -0.07[**] | 0.07[**] |
| | (-0.06, -0.02) | (0.04, 0.08) | (-0.02, 0.02) | (0.00, 0.05) | (-0.12, -0.03) | (0.02, 0.12) |
| *> $75,000* vs. *above poverty ≤ $75,000* | -0.01 | 0.01 | -0.02[*] | 0.00 | -0.03 | 0.02 |
| | (-0.03, 0.00) | (-0.01, 0.02) | (-0.03, 0.00) | (-0.01, 0.02) | (-0.07, 0.00) | (-0.02, 0.06) |
| Mother's education | | | | | | |
| High school *vs. less than high school* | 0.08[***] | -0.07[***] | 0.04[*] | -0.06[**] | 0.08[**] | -0.04 |
| | (0.05, 0.11) | (-0.1, -0.05) | (0.01, 0.06) | (-0.09, -0.02) | (0.02, 0.13) | (-0.10, 0.02) |
| Some college *vs. less than high school* | 0.11[***] | -0.11[***] | 0.05[***] | -0.08[***] | 0.05 | -0.08[*] |
| | (0.08, 0.13) | (-0.14, -0.08) | (0.02, 0.08) | (-0.12, -0.05) | (0.00, 0.11) | (-0.14, -0.02) |
| College degree *vs. less than high school* | 0.09[***] | -0.12[***] | 0.03[*] | -0.08[***] | 0.02 | -0.08[**] |
| | (0.06, 0.12) | (-0.15, -0.09) | (0.00, 0.06) | (-0.11, -0.04) | (-0.03, 0.08) | (-0.14, -0.02) |
| Census region[2] | | | | | | |
| Midwest *vs. Northeast* | 0.02[*] | -0.01 | 0.00 | -0.01 | 0.00 | -0.02 |
| | (0.00, 0.04) | (-0.03, 0.01) | (-0.02, 0.01) | (-0.03, 0.01) | (-0.04, 0.05) | (-0.07, 0.02) |
| South *vs. Northeast* | 0.01 | 0.00 | -0.02[*] | -0.01 | -0.01 | 0.01 |
| | (0.00, 0.03) | (-0.02, 0.01) | (-0.04, 0.00) | (-0.03, 0.01) | (-0.05, 0.03) | (-0.04, 0.05) |
| West *vs. Northeast* | 0.01 | -0.01 | 0.00 | 0.01 | 0.03 | -0.04 |
| | (-0.01, 0.03) | (-0.03, 0.01) | (-0.03, 0.02) | (-0.01, 0.04) | (-0.02, 0.09) | (-0.09, 0.01) |
| Not vaccinated against…[2] | | | | | | |
| Tdap & dependent variable vaccine *vs. only primary vaccine* | 0.00 | 0.03 | 0.07[***] | -0.06[***] | -- | -- |
| | (-0.05, 0.05) | (-0.02, 0.08) | (0.04, 0.09) | (-0.09, -0.02) | | |
| MenACWY & dependent variable vaccine *vs. only primary vaccine* | 0.04[***] | -0.02[*] | -- | -- | -0.06[*] | -0.25[***] |
| | (0.03, 0.06) | (-0.03, 0.00) | | | (-0.11, -0.01) | (-0.31, -0.20) |

*(Continued)*

**Table 3.** (Continued)

| | HPV AME (95% CI) | | MenACWY AME (95% CI) | | Tdap AME (95% CI) | |
|---|---|---|---|---|---|---|
| | *Agentic Refusal* | *Resource Failure* | *Agentic Refusal* | *Resource Failure* | *Agentic Refusal* | *Resource Failure* |
| HPV & dependent variable vaccine *vs. only primary vaccine* | -- | -- | 0.10[***] | -0.13[***] | 0.09[**] | -0.11[***] |
| | | | (0.08, 0.11) | (-0.15, -0.11) | (0.04, 0.14) | (-0.17, -0.05) |
| HPV, MenACWY & Tdap *vs. only primary vaccine* | 0.06[***] | -0.03[***] | 0.17[***] | -0.17[***] | 0.11[***] | -0.29[***] |
| | (0.04, 0.08) | (-0.05, -0.02) | (0.15, 0.19) | (-0.19, -0.14) | (0.07, 0.16) | (-0.34, -0.24) |
| State vaccine mandate | -0.02 | 0.00 | 0.00 | -0.03[***] | 0.00 | -0.03 |
| | (-0.07, 0.03) | (-0.05, 0.05) | (-0.01, 0.02) | (-0.05, -0.01) | (-0.04, 0.04) | (-0.07, 0.01) |
| Predicted probability base | 0.40 | 0.34 | 0.23 | 0.57 | 0.34 | 0.44 |
| N | 87,163 | | 54,726 | | 10,947 | |

[a]95% confidence intervals in parentheses.

[b]All comparisons not shown.

likely to be unvaccinated as a result of agentic refusal reasons compared to females (HPV AME = -0.07, CI = -0.08, -0.06; MenACWY AME = -0.03, CI = -0.04, -0.02; Tdap AME = -0.04, CI = -0.07, 10.01). However, while males were more likely to be unvaccinated as a result of resource failure reasons in the HPV and MenACWY samples (HPV AME = 0.09, CI = 0.08, 0.10; MenACWY AME = 0.03, CI = 0.02, 0.05), there was no notable difference in the probability of being unvaccinated as a result of resource failure reasons by teen's sex in the Tdap sample (AME = 0.01, CI = -0.02, 0.04).

Racial and ethnic differences in the reasons why teens were unvaccinated were inconsistent, but nevertheless present, across vaccine type. Hispanic, Black, and multiracial/other race teens were more likely to be unvaccinated as a result of resource failure reasons and less likely to be unvaccinated as a result of agentic refusal reasons in the HPV sample. There were no racial/ethnic differences in the predicted probability of the reason why teens are unvaccinated against MenACWY, with one exception being that multiracial/other race teens are more likely to be unvaccinated as a result of resource failure reasons compared to White teens (AME = 0.03, CI = 0.01, 0.06). Within the Tdap sample, the primary racial/ethnic differences in the predicted probability of resource reasons centered around Hispanic teens. Hispanic teens had a 0.09 higher predicted probability of being unvaccinated due to resource failure reasons compared to White teens (CI = 0.04, 0.14) and a 0.08 higher predicted probability of being unvaccinated due to resource failure reasons compared to Black teens (CI = 0.02, 0.14).

Similarly, family SES was inconsistently associated with agentic refusal and resource failure. The relationship between family SES and reasons why teens are unvaccinated was strongest in the HPV sample. Compared to teens whose family incomes are below the poverty line, teens living in higher income households were substantially less likely to be unvaccinated against HPV as a result of resource failure reasons (income > $75,000 AME = 0.05, CI = 0.03, 0.07; income < $75,000 AME = 0.06, CI = 0.04, 0.08). Furthermore, compared to teens whose mothers had less than a high school degree, teens whose mothers have a high school degree or greater are more likely to be unvaccinated against HPV due to agentic refusal reasons (high school AME = 0.08, CI = 0.05, 0.11; some college AME = 0.11, CI = 0.08, 0.13; college degree AME = 0.09, CI = 0.06, 0.12).

Contrastingly, there were no notable differences in the predicted probabilities of the reason why teens are unvaccinated in the MenACWY vaccine by family income. Meanwhile, mother's education was consistently associated with higher predicted probabilities of agentic refusal for

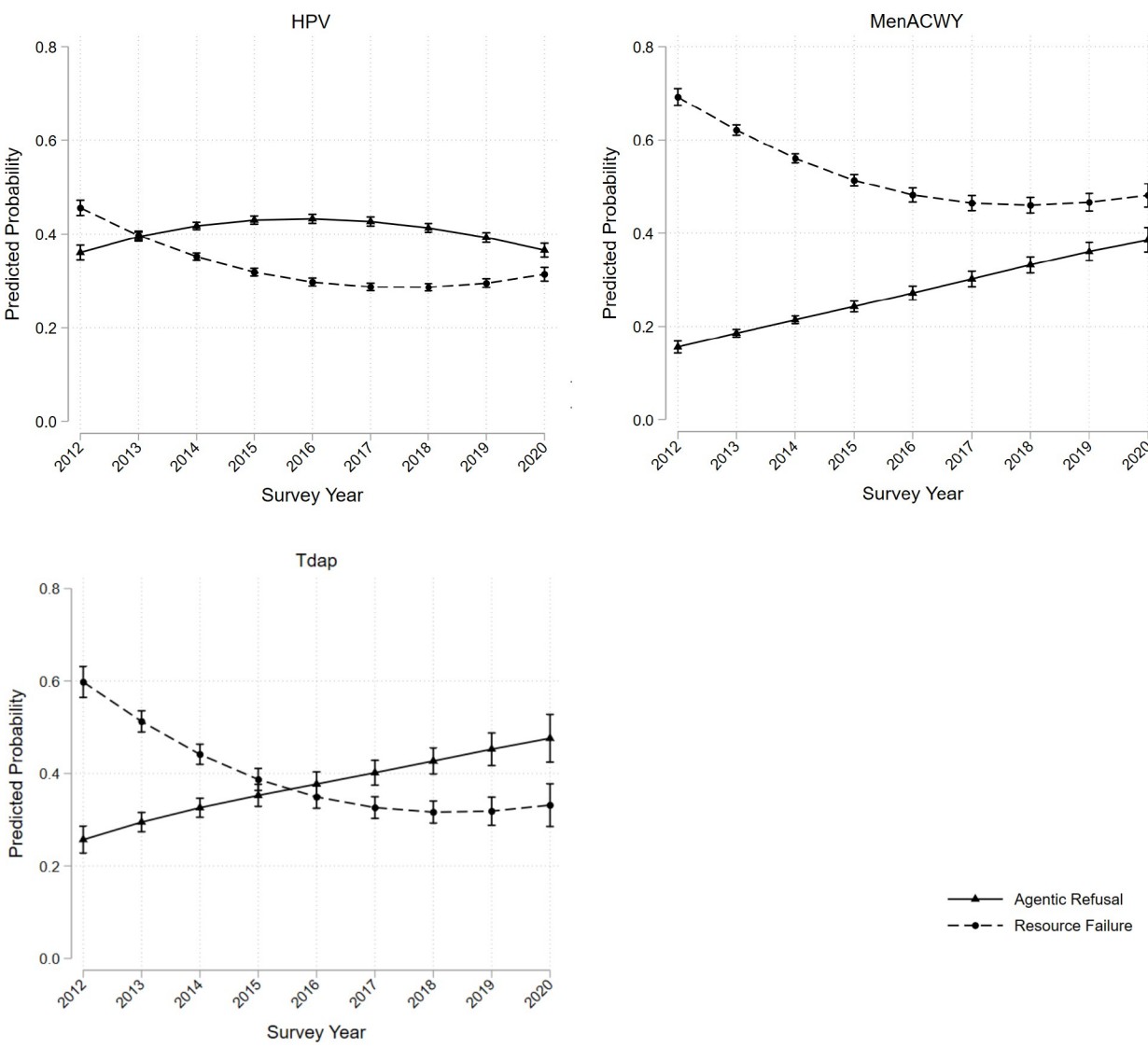

**Fig 1. Predicted probabilities for resource failure and agentic refusal reasons for why teens are unvaccinated.** HPV, MenACWY, and Tdap samples, NIS-Teen 2012–2020.

teens in the MenACWY sample. Compared to teens whose mothers had less than a high school degree, teens whose mothers had higher levels of education were consistently more likely to be unvaccinated against MenACWY as a result of agentic refusal reasons and less likely to be unvaccinated as a result of resource failure reasons (high school AME = -0.06, CI = -0.09, -0.02; some college AME = -0.08, CI = -0.12, -0.05; college degree AME = -0.08, CI = -0.11, -0.04).

Finally, family SES was unevenly associated with the reasons why teens are unvaccinated against Tdap. There was not a substantial difference in the predicted probability of resource failure or agentic refusal reasons for why teens are unvaccinated against Tdap when teens whose family with incomes below the poverty line are compared to teens whose family incomes exceeded $75,000. However, compared to teens whose family incomes were between the poverty line and $75,000, teens with family incomes below the poverty line were less likely to be unvaccinated against Tdap as a result of agentic refusal reasons (agentic refusal AME = -0.07, CI = -0.12, -0.03) and more likely to be unvaccinated as a result of resource failure

reasons (AME = 0.07, CI = 0.02, 0.12). Unlike teens in the HPV and MenACWY samples, maternal education was inconsistently associated with the predicted probability of reasons why teens are unvaccinated against Tdap. The only substantial difference in the predicted probability of agentic refusal reasons why teens are unvaccinated occurred in the comparison of teens whose mothers have a high school degree compared to those who had less than a high school degree (AME = 0.08, CI = 0.02, 0.13). Meanwhile, teens whose mothers had some college education or a college degree were no more likely to be unvaccinated against Tdap as a result of agentic refusal reasons compared to teens whose mothers had less than a college education. However, teens whose mothers had at least some college education or a college degree were less likely to be unvaccinated against Tdap due to resource failure reasons compared to teens whose mothers had less than a high school degree (some college AME = -0.08, CI = -0.14, -0.02; college degree AME = -0.08, CI = -0.14, -0.02).

Lastly, several patterns were evident when examining teen's vaccination status holistically. Across all three samples, teens missing all three adolescent vaccines were more likely to be unvaccinated as a result of agentic refusal reasons (HPV AME = 0.60, CI = 0.04, 0.08; MenACWY AME = 0.17, CI = 0.15, 0.19; Tdap AME = 0.11, CI = 0.07, 0.16) and less likely to be unvaccinated as a result of resource failure reasons (HPV AME = -0.03, CI = -0.05, -0.02; MenACWY AME = -0.17, CI = -0.19, -0.14; Tdap AME = -0.29, CI = -0.34, -0.24). Lastly, state mandates for each of the teen vaccines did not substantially impact the predicted probability of teens being unvaccinated as a result of agentic refusal reasons. Notably, state mandates only meaningfully affected the predicted probability of being unvaccinated as a result of resource failure reasons for teens in the MenACWY sample (AME = -0.03, CI = -0.05, -0.01) and did not have a similar effect in the HPV and Tdap samples.

## Discussion

The quantitative identification of disparities implicitly rests on the assumption that everyone is acting to access healthcare in line with public health and clinical recommendations [e.g., Healthy People 2030, 56]. However, trends whereby privileged groups activate resources to achieve care that goes against clinical guidance have the potential to obscure inequities in access. This study investigates this possibility through an examination of the reasons why teens are unvaccinated and by examining when and how inequities in resources are relevant in the reasons why teens are unvaccinated. The finding that resource failures are associated with a substantial portion of the reasons why teens are unvaccinated and tends to disproportionately affect racial/ethnic minority and low SES teens, even in the context of the absence of similar disparities in rates of vaccination makes novel contributions both to vaccination research and health inequalities more broadly.

This study contributes to a growing body of literature that calls for an examination of the underlying mechanisms of inequity in the absence of inequalities in rates [57]. A focus on inequalities in resources is one fruitful avenue through which we can do so. As decades of research have demonstrated [1], inequalities in health and healthcare are persistent and challenging to address due to tenacious disparities in access to health-promoting resources. Consequently, scholars should continue to approach the absence of inequalities with a critical lens.

Methodologically, these findings illustrate the potential pitfalls of relying on population-level rates as a measure of equity. Because of the durable nature of inequalities, an examination of rates of uptake should only be the initial step in the investigation of health disparities. Qualitative research is well suited to identify the mechanisms underlying health disparities. However, quantitative researchers can also find creative ways to identify the mechanisms that generate inequities as well. As this research shows, one way to do this is through an

examination of the reasons why individuals have not accessed a service, thereby enabling the identification of systematic barriers to care. Researchers studying health should critically examine the reliance on rates as a measure of health equity and consider the inclusion of other measures that capture structural barriers in their efforts to capture health inequities, not simply health inequalities.

This research makes several contributions to research on vaccination patterns in the United States. While previous studies have compared the county-level SES of counties with low-levels of vaccination [14], examined the characteristics of children unvaccinated against all vaccines compared to only select vaccines [13], and examined the frequency of a subset of the reasons why teens are unvaccinated using the NIS-Teen [58, 59], this study is the first to leverage a broad scope of the reasons why teens are unvaccinated to operationalize differing sets of resources. By using these reasons for why teens are unvaccinated to represent sets of resources, this study is able to directly compare the social characteristics of the parents of and the teens who are unvaccinated as a result of vaccine hesitancy or undervaccination reasons. Consequently, this study can demonstrate the relative importance of resources in driving the reasons why teens are unvaccinated.

These findings indicate the need to re-focus on addressing barriers to vaccination initiation among teens and complements other findings that disparities in vaccine series completion [7, 60]. Although the relative importance of resource failure reasons as the reason why teens are unvaccinated has decreased over time across all three vaccines, resource failure reasons explained a notable portion of reasons why teens were unvaccinated. Furthermore, although there were not consistent racial/ethnic and socioeconomic disparities in the probability of resource failure reasons for why teens are unvaccinated across all three samples, important disparities remained, particularly in the HPV sample. The inability to access resources that would allow parents to overcome obstacles to vaccination continues to be a substantial issue that further research and policy interventions should address, and the absence of traditional sociodemographic disparities in rates of vaccination should not inhibit future efforts to alleviate inequities.

Although structurally disadvantaged groups have greater access to health-promoting resources, the rising tide of vaccine hesitancy among structurally advantaged groups likely contributes to their lower rates of adolescent vaccine initiation [12, 13]. Nevertheless, addressing the continued barriers to vaccination is important even as the rate of initiation for adolescent vaccines is higher for some structurally disadvantaged groups [6, 7]. Notably, the WHO only includes vaccine hesitancy - not barriers to vaccination - as one of the ten threats to global health [15]. This research shows that public health efforts to increase the rates of vaccination must take a two-pronged approach that both works to address vaccine hesitancy as well as the structural barriers that inhibit vaccination. This could include increased efforts to provide vaccination clinics in schools and community centers as well as including information about vaccine recommendations within larger efforts to address vaccine hesitancy.

## Limitations

This study's findings must be interpreted within several limitations. The study categorizes a broad spectrum of reasons why teens are unvaccinated into three categories, which necessarily simplifies decisions which are inherently complex. The advantage of this simplification is that it allows for a broad understanding of patterns, but it necessarily eliminates much of the intricacy of vaccination decision-making [17], and there is likely heterogeneity in the resources activated within each category. For example, a study of social media posts and online discussion found that Black mothers feel as if they have less freedom to opt out of vaccines compared

to White mothers due to the increased state surveillance on Black families [43]. Nevertheless, despite this study's simplification of vaccination decisions, its investigation of broad level reasons for why teens are unvaccinated and can guide future research.

Additionally, there are two potential populations that are not included in the NIS-Teen subset of parents who answered the question of why their teen was unvaccinated. First, parents who reported that they were likely to vaccinate their child against HPV in the next year were not asked why their teen was unvaccinated, even though parents whose teens were unvaccinated against the MenACWY and Tdap vaccine were not similarly excluded. Consequently, this survey does not capture some parents who intend to vaccinate their teen against HPV but face barriers to vaccination, which could lead to an underestimation of the role of resource failure reasons that constrain vaccination decisions. Second, this sample does not capture all parents who did not vaccinate their child, because parents are likely to recall that their child was vaccinated, even when medical records indicate that the child is not up to date, which could underestimate the number of unvaccinated children [61]. Despite these sample limitations, this subset of parents from a large nationally representative sample of parents gives us insight into the broad forces that constrain or enable vaccination decisions.

## Conclusion

This study's investigation of the heterogenous resources underlying teen's unvaccinated status generates several important implications, both for the investigation of both the specific study of vaccination decisions and the broader study of resources and healthcare inequalities. Regarding decisions not to vaccinate, this study calls for increased attention to the barriers that socially marginalized parents face in accessing vaccination for their children. Even as sociodemographic disparities in vaccination rates decline, unequal access to resources continues to drive inequity. While there are a substantial proportion of parents who are making active decisions against vaccinating their children, an exclusive focus on these parents inhibits our ability to address health equity by facilitating vaccination for all children.

## Supporting information

**S1 Table. Sensitivity analyses, multinomial logistic regression models predicting reason for teen's unvaccinated status for the HPV, MenACWY, and Tdap samples, NIS-Teen 2012–2020.**
(DOCX)

**S2 Table. Average marginal effects for the multinomial logistic regressions predicting other reason for teen's unvaccinated status for the HPV, MenACWY, and Tdap samples, NIS-Teen 2012–2020.**
(DOCX)

**S1 File. State vaccine mandate data.**
(DOCX)

**S2 File. U.S.A. state-level vaccine mandate measure.**
(XLSX)

## Acknowledgments

I am grateful for the generous feedback provided by Brea Perry, Jessica Calarco, Emily Ekl, Brian Powell, Andrew Halpern-Manners, Caroline Brooks, and Robert Gallagher through each iteration of this project.

## Author Contributions

**Conceptualization:** Elizabeth M. Anderson.

**Data curation:** Elizabeth M. Anderson.

**Formal analysis:** Elizabeth M. Anderson.

**Investigation:** Elizabeth M. Anderson.

**Methodology:** Elizabeth M. Anderson.

**Software:** Elizabeth M. Anderson.

**Visualization:** Elizabeth M. Anderson.

**Writing – original draft:** Elizabeth M. Anderson.

**Writing – review & editing:** Elizabeth M. Anderson.

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
