## [Decision Letter · Decision Letter 0]

6 Sep 2023

PONE-D-23-15160Obscured Inequity: How Focusing on Rates of Disparities Can Conceal Inequities in The Reasons Why Adolescents Are UnvaccinatedPLOS ONE

Dear Dr. Anderson,

Thank you for submitting your manuscript to PLOS ONE. After careful consideration, we feel that it has merit but does not fully meet PLOS ONE’s publication criteria as it currently stands. Therefore, we invite you to submit a revised version of the manuscript that addresses the points raised during the review process.

We look forward to receiving your revised manuscript.

Kind regards,

Anat Gesser-Edelsburg, Ph.D.

Academic Editor

PLOS ONE

Journal Requirements:

Reviewers' comments:

Reviewer's Responses to Questions

**Comments to the Author**

1. Is the manuscript technically sound, and do the data support the conclusions?

Reviewer #1: Yes

Reviewer #2: Yes

2. Has the statistical analysis been performed appropriately and rigorously? 

Reviewer #1: Yes

Reviewer #2: I Don't Know

3. Have the authors made all data underlying the findings in their manuscript fully available?

Reviewer #1: Yes

Reviewer #2: Yes

4. Is the manuscript presented in an intelligible fashion and written in standard English?

Reviewer #1: Yes

Reviewer #2: Yes

5. Review Comments to the Author

Reviewer #1: The article is of excellent quality. It is a very important topic for the scientific community. The small points I suggest improving are:

- Present an international overview of the theme in the introduction (what was done similarly in other countries, or even what was not done);

- Improve the practical implications of the study (How important is the study for society? How important is the study for professionals? How important is the study for the health system? Etc..)

Reviewer #2: Well done. A few minor revisions needed. Overall, this paper is well thought, thorough and important in addressing health disparities. I applaud the author for studying undervaccination and vaccine hesitancy together. This is very critical. The author has done an incredible job in clarifying terms. However, some definitions may need further explanation rather than broad classification. Not sure if this would be beyond the scope of the work, but it would be great to unpack “vaccine hesitancy/ refusal” spectrum since there is evidence that some vaccine hesitant parents may still vaccinate their children. Similarly, with “undervaccination”, does the author mean ‘incomplete dosage’, ‘untimely vaccinations’, or both, given that the author uses data on unvaccinated/ partially vaccinated adolescents. This is an excellent paper that can be further strengthened by adding some information in the discussion section.See attached document for specific details

6. PLOS authors have the option to publish the peer review history of their article (what does this mean?). If published, this will include your full peer review and any attached files.

Reviewer #1: **Yes: **Mateus Dias Antunes

Reviewer #2: No

---

## [Decision Letter · Decision Letter 1]

23 Oct 2023

Obscured Inequity: How Focusing on Rates of Disparities Can Conceal Inequities in The Reasons Why Adolescents Are Unvaccinated

PONE-D-23-15160R1

Dear Dr. Anderson,

We’re pleased to inform you that your manuscript has been judged scientifically suitable for publication and will be formally accepted for publication once it meets all outstanding technical requirements.

Kind regards,

Anat Gesser-Edelsburg, Ph.D.

Academic Editor

PLOS ONE

Additional Editor Comments (optional):

Reviewers' comments:

Reviewer's Responses to Questions

**Comments to the Author**

1. If the authors have adequately addressed your comments raised in a previous round of review and you feel that this manuscript is now acceptable for publication, you may indicate that here to bypass the “Comments to the Author” section, enter your conflict of interest statement in the “Confidential to Editor” section, and submit your "Accept" recommendation.

Reviewer #1: All comments have been addressed

Reviewer #2: All comments have been addressed

2. Is the manuscript technically sound, and do the data support the conclusions?

Reviewer #1: Yes

Reviewer #2: Yes

3. Has the statistical analysis been performed appropriately and rigorously? 

Reviewer #1: Yes

Reviewer #2: I Don't Know

4. Have the authors made all data underlying the findings in their manuscript fully available?

Reviewer #1: Yes

Reviewer #2: Yes

5. Is the manuscript presented in an intelligible fashion and written in standard English?

Reviewer #1: Yes

Reviewer #2: Yes

6. Review Comments to the Author

Reviewer #1: The authors made all the adjustments that were requested in the first evaluation. I suggest accepting the article now.

Reviewer #2: Excellent work. All comments were addressed appropriately and completely. Looking forward to reading the published manuscript.

7. PLOS authors have the option to publish the peer review history of their article (what does this mean?). If published, this will include your full peer review and any attached files.

Reviewer #1: **Yes: **Mateus Antunes

Reviewer #2: No

---

## [Editor Report · Acceptance letter]

25 Oct 2023

PONE-D-23-15160R1 

Obscured inequity: How focusing on rates of disparities can conceal inequities in the reasons why adolescents are unvaccinated 

Dear Dr. Anderson:

I'm pleased to inform you that your manuscript has been deemed suitable for publication in PLOS ONE. Congratulations! Your manuscript is now with our production department. 

Kind regards, 

on behalf of

Prof. Anat Gesser-Edelsburg 

Academic Editor

PLOS ONE